# GraspGF: Learning Score-based Grasping Primitive for Human-assisting Dexterous Grasping

**Tianhao Wu** [1,2,3*], **Mingdong Wu** [1,3*], **Jiyao Zhang**[1,2,3], **Yunchong Gan**[1], **Hao Dong**[1,3 †]

[1] Center on Frontiers of Computing Studies, School of Computer Science, Peking University
[2] Beijing Academy of Artificial Intelligence
[3] National Key Laboratory for Multimedia Information Processing,
School of Computer Science, Peking University
{thwu,jiyaozhang}@stu.pku.edu.cn, {wmingd,yunchong,hao.dong}@pku.edu.cn

## Abstract

The use of anthropomorphic robotic hands for assisting individuals in situations where human hands may be unavailable or unsuitable has gained significant importance. In this paper, we propose a novel task called *human-assisting dexterous grasping* that aims to train a policy for controlling a robotic hand's fingers to assist users in grasping objects. Unlike conventional dexterous grasping, this task presents a more complex challenge as the policy needs to adapt to diverse user intentions, in addition to the object's geometry. We address this challenge by proposing an approach consisting of two sub-modules: a hand-object-conditional grasping primitive called **Grasp**ing **G**radient **F**ield (GraspGF), and a history-conditional residual policy. GraspGF learns 'how' to grasp by estimating the gradient from a success grasping example set, while the residual policy determines 'when' and at what speed the grasping action should be executed based on the trajectory history. Experimental results demonstrate the superiority of our proposed method compared to baselines, highlighting user awareness and practicality in real-world applications. The codes and demonstrations can be viewed at https://sites.google.com/view/graspgf.

## 1 Introduction

The significance of human hands in everyday life cannot be overstated. However, there are situations where they may not always be available, especially in scenarios where an individual may have upper limb loss or need to interact with hazardous objects. In such instances, utilising an anthropomorphic dexterous robotic hand for assistance can be a viable option [1]. Such a dexterous hand possesses a high degree of freedom, allowing it to handle diverse daily tasks [2, 3, 4], given that many everyday objects are designed to match the structure of the human hand. This inspired us to propose a novel task called *human-assisting dexterous grasping*, in which a policy is trained to assist users with upper limb loss in grasping objects by controlling the robotic hand's fingers, as illustrated in Figure 1a.

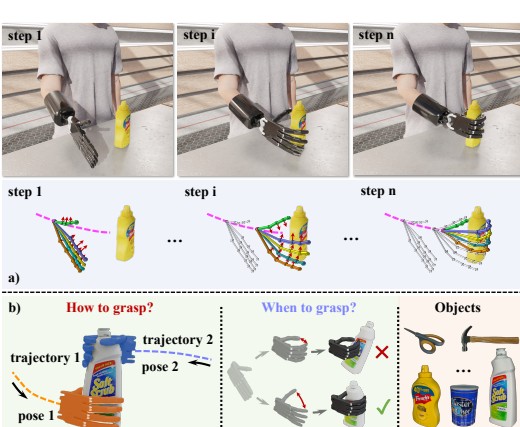

**Figure 1: a)** Demonstration of human-assisting dexterous grasping. **b)** Challenges of our setting.

Traditional teleoperation methods [5, 6] are unsuitable for assisting upper limb amputees in grasping because no information about the human fingers can be accessed. Compared to conventional dexterous

---

*: equal contribution, †: corresponding author

37th Conference on Neural Information Processing Systems (NeurIPS 2023).

grasping, human-assisting dexterous grasping poses a more complex challenge, as the policy must adapt to an exponentially growing number of pre-conditions. As shown in Figure 1b, human users may grasp an object with various intentions, such as grasping different parts for different purposes or moving the hand and wrist at different speeds due to the complexity and diversity of human behaviour. Consequently, the conditional policy must be tailored not only to the object's geometry, as required by conventional dexterous grasping, but also to the user's intentions, requiring the policy to be *user-aware*. In this context, open-looped methods such as grasp pose generation [7, 8, 9, 10] and classification-based methods [11, 12, 13] may fall short, as they do not factor in the user's intention. Reinforcement learning (RL) [2, 14, 15] presents a natural solution by enabling the training of a closed-loop human-object-conditioned policy. However, in human-assisting dexterous grasping, RL may encounter more severe generalisation issues, due to the need to generalise to diverse grasping pre-conditions. Prior RL-based approaches [16, 17] have explored leveraging human-collected and engineering-heavy demonstrations to address this issue. However, collecting a large volume of diverse demonstrations that encompass different objects, grasping timings, and locations may not be feasible.

To address the challenges associated with achieving dexterous grasping for assisting humans, an effective policy needs to tackle the following two crucial questions:

> *1) **How** should the robot grasp the object considering the current relative pose between the user and the object? 2) **When** and at what speed should the robot execute the grasping action based on the user movement trajectory history?*

In this paper, we present a novel approach that consists of two sub-modules designed to address the aforementioned questions individually: 1) a hand-object-conditional grasping primitive, and 2) a history-conditional residual policy. The grasping primitive, which we call ***Grasping Gradient Field (GraspGF)***, is trained to learn '***how** to grasp*' by estimating the score function, *i.e.*, the gradient of the log-density, of a success grasping example set. The GraspGF outputs a gradient that indicates the fastest direction to increase the 'grasping likelihood' conditioned on an object and the user's wrist. The gradient can be translated into primitive controls on each finger joint, enabling the fingers to reach an appropriate grasp pose iteratively. However, GraspGF is not capable of determining how fast the fingers should move along the gradient as it is history-agnostic. To determine '***when** to grasp*', we train a residual policy that outputs a 'scaling action' that determines how fast the finger joints should move along with the primitive action, based on the history of the wrist's trajectory. Besides, as the primitive policy is not aware of the environment dynamics due to the offline training, we further require the residual policy to output a 'residual action' to correct the primitive action.

Our proposed approach offers several conceptual advantages. GraspGF leverages the strong conditional generative modelling of score-based methods, as demonstrated in prior works [18, 19, 20], enabling it to output promising primitive actions conditioned on novel user's intentions. Additionally, the residual-learning design of GraspGF facilitates cold start exploration and enhances the efficiency of residual policy training. Compared to demonstration-based methods, our approach only requires a synthesised grasping example set and does not rely on exhaustive human labelling or extensive engineering effort, making it more practical for implementation in real-world applications.

In our experiments, we evaluate several methods on a dexterous grasping environment that assists humans in grasping over 4900+ on-table objects with up to 200 realistic human wrist movement patterns. Our comparative results demonstrate that our proposed method significantly outperforms the baselines across various metrics. Ablation studies further confirm the effectiveness of our proposed grasping gradient field and residual-learning design. Our analysis reveals that our method's superiority lies in its user-awareness, *i.e.*, our trained policy is more tailored to the user's intentions. Additionally, we conduct real-world experiments to validate the practicality of our approach. Our results indicate that our trained model can generalise to the real world to some degree without fine-tuning.

Our contributions are summarized as follows:

- We introduce a novel and challenging human-assisting dexterous grasping task, which may potentially help social welfare.

- We propose a novel two-stage framework that decomposes the task into learning a primitive policy via score-matching and training a residual policy to complement the primitive policy via RL.

- We conduct experiments to demonstrate our method significantly outperforms the baselines and the effectiveness of our method deployed in the real world.

## 2 Human-assisting Dexterous Grasping

We study the *human-assisting dexterous grasping*, in which a policy is trained to assist users in grasping objects by controlling the robotic hand's fingers. We formulate the problem as follows:

**State and Action Spaces:** In this task, we consider a human-assisting grasping scenario involving a 28-DoF 5-fingered robotic hand. The 18-DoF joints of the fingers are denoted as $\mathbf{J} \in \mathbb{R}^{18}$, the 4-DoF under-actuated joints of the fingers are denoted as $\mathbf{J}^u \in \mathbb{R}^4$, and the 6-DoF pose of the wrist is represented by $\mathbf{b} = [\mathbf{b}_p, \mathbf{b}_q]$, where $\mathbf{b}_p \in \mathbb{R}^3$ denotes the 3-D position and $\mathbf{b}_q \in \mathbb{R}^4$ represents the 4-D quaternion. The action space $\mathcal{A} \subseteq \mathbb{R}^{18}$ corresponds to the 18-D relative changes applied to the hand joints. Unlike traditional dexterous grasping tasks, the action space does not include the 6-D relative changes for the wrist, since the wrist pose is controlled by a human user.

**Task Simulation:** To simulate the movement of the human user's wrist, we sample a wrist trajectory at the start of each episode $\tau_{\mathbf{b}} = \{\mathbf{b}_1, \mathbf{b}_2, ..., \mathbf{b}_T\}$ ($T$ denotes the horizon). At each time step $t$, the wrist's pose is set to $\mathbf{b}_t$. Specifically, we initially sample a target object $O \sim p_O(O)$ from an object prior distribution. Then, the wrist trajectory is sampled $\tau_{\mathbf{b}} \sim p_{\tau_{\mathbf{b}}}(\tau_{\mathbf{b}}|O)$ conditioned on the target object $O$. The terminal wrist state, $\mathbf{b}_T$, is designed to be 'graspable'. In other words, there exists a feasible hand joint $\mathbf{J}^*$ such that the grasp pose $[\mathbf{J}^*, \mathbf{b}_T]$ can successfully grasp the object $O$.

**Observations:** This task requires the agent to adapt to both the wrist trajectories $\tau_{\mathbf{b}}$ and different objects $O$. Consequently, the policy $\pi(\mathbf{a}|\cdot)$ should be conditioned on the finger joints $\mathbf{J}$, visual observations $o$, and the history of hand wrist poses $H_t = [\mathbf{b}_{t-k}, \mathbf{b}_{t-k+1}, ..., \mathbf{b}_t]$, where $k$ is a hyper-parameter. In this work, we use the full point cloud of the target object as the visual observation $o$ and consider the last five wrist states as the history $H_t = [\mathbf{b}_{t-4}, ..., \mathbf{b}_t]$. Note that the RL-based policies used in all the experiments also take under-actuated joints $\mathbf{J}^u \in \mathbb{R}^4$ as input, for simplicity, we omit this input in the following notations.

**Objective:** The objective is to find a policy $\pi(\mathbf{a}|\mathbf{J}, o, H)$ that maximizes the expected grasping success rate over the initial distributions, i.e. $O \sim p_O(O)$ and $\tau_{\mathbf{b}} \sim p_{\tau_{\mathbf{b}}}(\tau_{\mathbf{b}}|O)$:

$$\pi^* = \arg\max_{\pi} \mathbb{E}_{\substack{O \sim p_O(O), \tau_{\mathbf{b}} \sim p_{\tau_{\mathbf{b}}}(\tau_{\mathbf{b}}|O), \\ \mathbf{a}_t \sim \pi(\cdot|\mathbf{J}_t, o_t, H_t)}} [\mathbb{1}(\text{success})] \tag{1}$$

Eq. 1 poses a challenging objective since the policy should generalize not only to different objects $O \sim p_O(O)$, as required in conventional dexterous grasping, but also to different hand wrist trajectories $\tau_{\mathbf{b}} \sim p_{\tau_{\mathbf{b}}}(\tau_{\mathbf{b}}|O)$. In other words, the agent should be *user-aware*.

## 3 Method

**Overview:** A user-aware policy needs to tackle the following two crucial questions: 1) ***How*** should the robot grasp the object considering the current relative pose between the user and the object? 2) ***When*** and at what speed should the robot execute the grasping action based on the user movement trajectory history? As illustrated in Figure 2, our key idea is to partition the task into two stages that address these questions individually: 1) Learning a primitive policy $\pi_p^{\theta}(\mathbf{a}_t^p|\mathbf{J}_t, o_t, \mathbf{b}_t)$ that proposes a primitive action $\mathbf{a}_t^p$ that can guide the fingers forming into a pre-grasp pose, from an success grasping pose example set. 2) Learning a residual policy $\pi_r^{\phi}(\mathbf{a}_t^s, \mathbf{a}_t^r|\mathbf{J}_t, o_t, H_t)$ that outputs a scaling action $\mathbf{a}_t$ to determine 'how fast' the joints should move with the primitive action and a residual action $\mathbf{a}_t^r$ that further corrects the overall action, via RL. The combined policy $\pi^{\theta,\phi}(\mathbf{a}_t|\mathbf{J}_t, o_t, H_t)$ is as follows:

$$\mathbf{a}_t^p \sim \pi_p^{\theta}(\mathbf{a}_t^p|\mathbf{J}_t, o_t, \mathbf{b}_t), \ (\mathbf{a}_t^s, \mathbf{a}_t^r) \sim \pi_r^{\phi}(\mathbf{a}_t^s, \mathbf{a}_t^r|\mathbf{J}_t, o_t, H_t)$$
$$\pi^{\theta,\phi}(\cdot|\mathbf{J}_t, o_t, H_t) = \mathbf{a}_t^p \odot \mathbf{a}_t^s + \mathbf{a}_t^r \tag{2}$$

Initially, we employ the score-matching to train the primitive policy $\pi_p^{\theta}$ from a grasping poses dataset. Subsequently, the combined policy, which is constructed from the residual policy $\pi_r^{\phi}$ is combined with the frozen $\pi_p^{\theta}$, and is trained under RL. In the following, we will introduce the motivations and the training procedures of the primitive policy (*i.e.*, GraspGF) and the residual policy in Sec 3.1 and Sec 3.2, respectively. The implementation details of both policies are described in Appendix B.

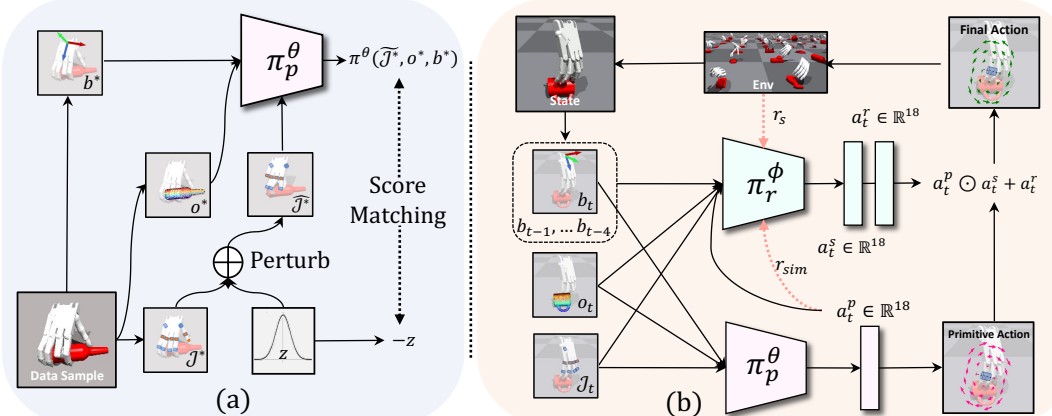

**Figure 2:** We decompose the human-assisting dexterous grasping into learning a primitive policy $\pi_p^\theta$ that learns to form a pre-grasp pose and a residual policy $\pi_r^\phi$ that learns to adjust the proceeding of the primitive action. **a)** The primitive policy $\pi_p^\theta$ is trained on success grasping examples via score-matching objective. **b)** The residual policy $\pi_r^\phi$ is trained to adjust the primitive policy via RL.

### 3.1 Learning GraspGF from Synthetic Examples

To address the first question above, we aim to search for a primitive policy $\pi_p^\theta$ that outputs action to maximize the likelihood of success, given a set of static conditions $(\mathbf{J}_t, o_t, \mathbf{b}_t)$. Inspired by [21], we can train such a policy by estimating the *score function* (*i.e.*, gradient of the log-density) of a conditional distribution $p_{\text{success}}(\mathbf{J}|o_t, \mathbf{b}_t)$:

$$\mathbf{a}^p = \pi_p^\theta(\cdot|\mathbf{J}, o, \mathbf{b}) = \nabla_{\mathbf{J}} \log p_{\text{success}}(\mathbf{J}|o, \mathbf{b}) \tag{3}$$

the $p_{\text{success}}(\mathbf{J}|o, \mathbf{b})$ denotes a fingers-joints' distribution that can successfully grasp the objects, given the current observation $o$ and the hand wrist $\mathbf{b}$. By definition, the score function of the distribution $\nabla_{\mathbf{J}} \log p_{\text{success}}(\mathbf{J}|o, \mathbf{b})$ indicates the fastest direction to increase the likelihood of the $p_{\text{success}}(\mathbf{J}|o, \mathbf{b})$. Intuitively, if the fingers move in the direction of $\nabla_{\mathbf{J}} \log p_{\text{success}}(\mathbf{J}|o, \mathbf{b})$, they will probably reach a feasible grasp-pose in the future with the success-likelihood $p_{\text{success}}(\mathbf{J}|o, \mathbf{b})$ increases. Hence, we formulate learning the primitive policy $\pi_p^\theta$ as estimating the gradient field of the log-success-likelihood $\nabla_{\mathbf{J}} \log p_{\text{success}}(\mathbf{J}|o, \mathbf{b})$, namely ***Grasp**ing **G**radient **F**ield (GraspGF)*.

Thanks to the Denoising Score Matching (DSM) [22], we can obtain a guaranteed estimation of the GraspGF $\nabla_{\mathbf{J}} \log p_{\text{success}}(\mathbf{J}|o, \mathbf{b})$ from a set of success examples $\mathcal{D}_{\text{success}} = \{(\mathbf{J}_i^*, o_i^*, \mathbf{b}_i^*)\}_{i=1}^N$ where the $\mathbf{J}_i^*$ is the feasible finger joints for grasping conditioned on $(o_i^*, \mathbf{b}_i^*)$. In the following, we first revisit the preliminaries of DSM and then introduce how to employ DSM to estimate the GraspGF.

**Denoising Score-Matching** Given a set of data-points $\{\mathbf{x}_i \sim p(\mathbf{x})\}_{i=1}^N$ from an unknown data distribution $p(\mathbf{x})$, the score-based generative model aims at estimating the *score function* of a data distribution $\nabla_{\mathbf{x}} \log p(\mathbf{x})$ via a *score network* $\mathbf{s}_\omega(\mathbf{x}) : \mathbb{R}^{|\mathcal{X}|} \to \mathbb{R}^{|\mathcal{X}|}$. During inference, a new sample is generated by the Langevin Dynamics, which is out of our interest.

To estimate $\nabla_{\mathbf{x}} \log p(\mathbf{x})$, the Denoising Score-Matching (DSM) [22] proposes a tractable objective by pre-specifying a noise distribution $q_\sigma(\widetilde{\mathbf{x}}|\mathbf{x})$, *e.g.*, $\mathcal{N}(0, \sigma^2 I)$, and train the score network to denoise the perturbed data samples:

$$\mathcal{L}(\omega) = \mathbb{E}_{\widetilde{\mathbf{x}} \sim q_\sigma, \mathbf{x} \sim p(\mathbf{x})} \left[ ||\mathbf{s}_\omega(\widetilde{\mathbf{x}}) - \nabla_{\widetilde{\mathbf{x}}} \log q_\sigma(\widetilde{\mathbf{x}}|\mathbf{x})||_2^2 \right] \tag{4}$$

where $\nabla_{\widetilde{\mathbf{x}}} \log q_\sigma(\widetilde{\mathbf{x}}|\mathbf{x}) = \frac{1}{\sigma^2}(\mathbf{x} - \widetilde{\mathbf{x}})$ are tractable for the Gaussian kernel. DSM guarantees that the optimal score network holds $\mathbf{s}_\omega^*(\mathbf{x}) = \nabla_{\mathbf{x}} \log p(\mathbf{x})$ for almost all $\mathbf{x}$.

**Employing DSM to Estimate GraspGF** To estimate the GraspGF $\nabla_{\mathbf{J}} \log p_{\text{success}}(\mathbf{J}|o, \mathbf{b})$, we employ the DSM in Eq 4, the training objective of the primitive policy $\pi_p^\theta$ is derived as follows:

$$\mathcal{L}(\theta) = \mathbb{E}_{\substack{\widetilde{\mathbf{J}} \sim q_\sigma(\widetilde{\mathbf{J}}|\mathbf{J}^*), \\ (\mathbf{J}^*, o^*, \mathbf{b}^*) \sim \mathcal{D}_{\text{success}}}} \left[ \left\| \pi_p^\theta(\cdot|\widetilde{\mathbf{J}}, o^*, \mathbf{b}^*) - \frac{\mathbf{J}^* - \widetilde{\mathbf{J}}}{\sigma^2} \right\|_2^2 \right] \tag{5}$$

The Eq 5 is the L2 distance between the output of the primitive policy and the *denoising direction* $\frac{\mathbf{J}^* - \widetilde{\mathbf{J}}}{\sigma^2}$, *i.e.*, a direction pointing from the perturbed joints $\widetilde{\mathbf{J}}$ to the original joints $\mathbf{J}^*$. Intuitively, this objective is forcing the primitive policy to denoise the current joints to regions where the fingers are more likely to grasp the object.

## 3.2 Training Residual Policy via Reinforcement Learning

The human-assisting dexterous grasping cannot be effectively addressed solely through the primitive policy $\pi_p^\theta$. Although $\pi_p^\theta$ predicts the fastest direction to form a pre-grasp pose, it fails to determine the appropriate 'velocities' at which the joints should move in that direction. If the fingers close too quickly or too slowly, the agent may struggle to grasp the object. Additionally, due to offline training, the primitive policy lacks awareness of the dynamics of the environment, leading to potential violations of physical constraints.

To overcome these limitations, we propose the training of a residual policy $\pi_r^\phi$, which complements the primitive policy. The role of $\pi_r^\phi$ is twofold: firstly, it outputs a scaling action $\mathbf{a}^s$ that adjusts the speed of the primitive action, and secondly, it produces a residual action $\mathbf{a}^r$ that corrects the final output action. Eq 2 demonstrates that $\mathbf{a}^s$ can be interpreted as 18-D 'pseudo-velocities' imposed on the finger joints. With outputs less than 1 for $\mathbf{a}^s$, the residual policy can decelerate the primitive action, whereas values greater than 1 accelerate it.

To effectively control the proceeding of the primitive action, the residual policy $\pi_r^\phi(\mathbf{a}_t^r | \mathbf{J}_t, o_t, H_t)$ takes into account the history of the wrist $H_t$ and the object's point cloud $o$ as inputs. This enables the policy to infer the agent's speed of approach towards the object. Furthermore, the policy network incorporates the current joint state $\mathbf{J}_t$ to infer how to correct the primitive action $\mathbf{a}^p$.

We employ Proximal Policy Optimization (PPO) [23] to search for a final policy $\pi^{\theta, \phi}$ that maximises the following objective, where the primitive policy's parameters $\theta$ are frozen during training:

$$J(\phi) = \mathbb{E}_{\substack{O \sim p_O(O), \tau_\mathbf{b} \sim p_{\tau_\mathbf{b}}(\tau_\mathbf{b}|O), \\ \mathbf{a}_t \sim \pi^{\theta, \phi}(\cdot | \mathbf{J}_t, o_t, H_t)}} \left[ \sum_{t=0}^{T} \gamma^t r_t \right] \tag{6}$$

where $\gamma > 0$ denotes a discounted factor and $r_t$ is the reward at time-step $t$. To encourage the policy to successfully grasp the object while leveraging the primitive action $\mathbf{a}_t^p$ for efficient exploration, we assign the following simple reward function for training:

$$r_t = (1 - d_t) \cdot (r_{\text{sim}} + r_{\text{h}}) + d_t \cdot \lambda_s, \ d_t = \mathbb{1}(\text{success}, t = T)$$
$$r_{\text{sim}} = \lambda_a \cdot \left\langle \frac{\mathbf{a}^p}{||\mathbf{a}^p||_2}, \mathbf{J}_t - \mathbf{J}_{t-1} \right\rangle, r_{\text{h}} = \lambda_h \cdot \Delta h \tag{7}$$

where $\lambda_a, \lambda_h, \lambda_s > 0$ are hyperparameters and $\Delta h$ represents the change in the height of the object's centre of gravity after lifting. The term $d_t$ rewards the agent if the final grasp pose can successfully lift the target object. The term $r_{\text{sim}}$ is an intrinsic reward that encourages the agent when the joints-change follows the direction of the primitive action $\mathbf{a}_t^p$. We defer the PPO's hyperparameters to Appendix B.

## 3.3 Implementation Details

**Primitive Policy Network** This module takes the hand joints $\mathbf{J} \in \mathbb{R}^{18}$, object point cloud $o \in \mathbb{R}^{3 \times 1024}$ and the hand wrist $\mathbf{b} \in \text{SE}(3)$ as input, we first project the point cloud into the hand wrist's coordinate $o_\mathbf{b}$. We encode $o_\mathbf{b}$ into a 512-D global feature by PointNet++ [24]. The hand joints $\mathbf{J}$ and the noise-condition $t \in [0, 1]$ are further encoded into 1024-D and 512-D features respectively. Concatenating the features together, we feed the concatenated feature into MLPs to obtain the 18-D output.

**Residual Policy Network** Taking the hand joints $\mathbf{J}_t \in \mathbb{R}^{18}$, under-actuated joints $\mathbf{J}_t^u \in \mathbb{R}^4$, visual observation $o_t \in \mathbb{R}^{3 \times 1024}$ and the hand wrists' trajectory $H_t = [\mathbf{b}_t, ..., \mathbf{b}_{t-4}]$, $\mathbf{b}_i \in \text{SE}(3)$ as input, this module use the PointNet [25]to encoder the point cloud. Similarly, we first obtain the global feature $f_{o_\mathbf{b}} \in \mathbb{R}^{512}$ of the projected point cloud. The primitive action $\mathbf{a}_t^p = \pi_p^\theta(\mathbf{J}_t, o_t, \mathbf{b}_t)$, hand joints $\mathbf{J}$ and the history $H$ are concatenated and further encoded into 512-D features. Concatenating the above features together, we feed the concatenated feature into MLPs to obtain the 36-D output.

# 4 Experiment Setups

## 4.1 Task Simulation

**Environment setup**: We created a simulation environment based on the ShadowHand environment in Isaac Gym [26], using the ShadowHand model from IBS [27]. The simulation environment enables parallel training on hundreds of environments. Each environment consists of an object placed on the ground and a hand that follows a pre-generated human wrist trajectory. The agent can only control the joints of the hand. The episode horizon is 50 steps, and each episode only terminates at the final step. Following a similar setting to IBS [27], when the episode terminates, part of the joints of each finger will automatically close by 0.1 radians, and then the wrist of the hand will be lifted by 1m.

**Grasping pose generation**: We created our success grasping pose based on the UniDexGrasp dataset [10]. We filtered the data to match our ShadowHand model's degrees of freedom. The dataset was split into three sets: training instances (3127 objects, 363,479 grasps), seen category unseen instances (519 objects, 2595 grasps), and unseen category instances (1298 objects, 6490 grasps).

**Human grasping wrist patterns**: To mimic real human grasping patterns, we resampled 200 real human grasping wrist trajectories from HandoverSim! [28], from which we extracted 200 wrist movement patterns. These patterns were split into 150 training patterns and 50 testing patterns.

The details are deferred to Appendix A.

## 4.2 Metrics

Following the DexVIP [29], we report the following three metrics: 1) **Success**: The object can be lifted more than 0.1m off the table, while the change in distance between the object and hand position after lifting is smaller than 0.05m. 2) **Posture**: The distance between the target human hand pose and the agent hand pose. It tells us how human-like the learned grasps are. 3) **Stability**: The translation and rotation changes of the object during the agent's grasping process. Translation is measured by the Euclidean distance between the final object position and the initial object position. Rotation is measured by the cosine distance between the x-axis of the initial object and the final object.

## 4.3 Baselines

We compare our method with the following RL-based methods. Note that all baselines are re-trained by taking the latest 5-frame wrist states as input. 1) **IBS** [27]: IBS explicitly compute rich contact information according to hand mesh and object mesh, which has good generalisation to objects and different contact situation. We modify the baseline to only output joint actions, and keep the reward the same as previous. 2) **PPO** [23]: We adopt PPO as ours pure RL baseline. For pure RL, we use the reward of fingertip distance (distance of fingertip to the closest point of object point cloud), $r_h$ and success reward. 3) **PPO (Goal)**: We randomly sample the goal which corresponds to the current object as additional input to the agent. We add another goal pose matching reward compared to PPO baseline. 4) **ILAD** [30]: We choose ILAD as our RL+Imitation learning baseline, which will first generate sub-optimal grasping trajectories according to example grasp pose, then use these demonstrations for imitation learning and RL. We regenerate the trajectories for ILAD training based on our grasp data. We use the same reward as the PPO baseline. The details are deferred to Appendix C.

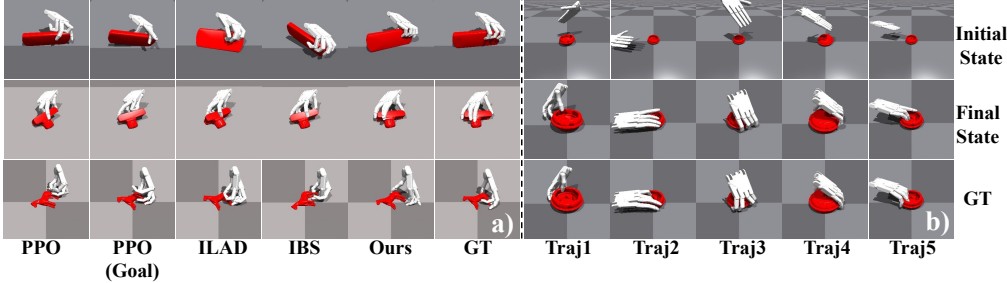

**Figure 3:** Qualitative results of comparison with baselines and different trajectories. a): final grasp poses of different methods. b): final grasp poses of our method under different human trajectories

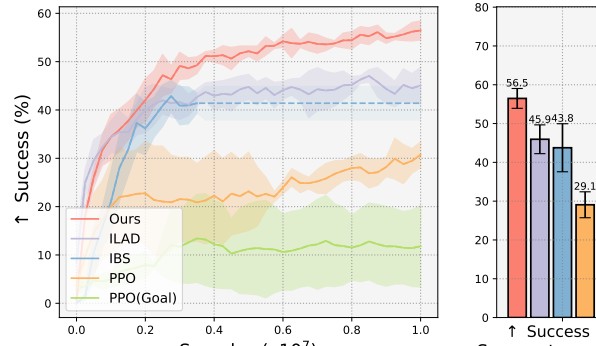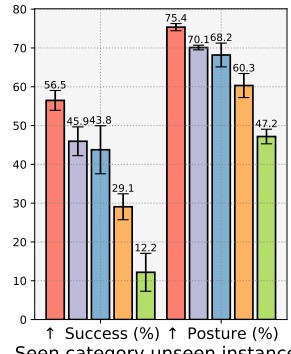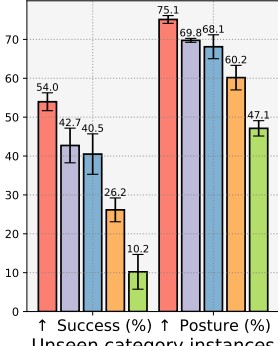

**Figure 4:** Quantitative comparative results. Left: training curve of different methods. Note that IBS takes 144 hours on V100 to reach 3.5 million agent steps, while ours only takes 15 hours to reach 10 million agent steps. Middle: Success and Posture of different methods on **seen category unseen instances**. Right: Success and Posture of different methods on **unseen category instances**.

## 5 Experimental Results

### 5.1 Comparative Results

As shown in Figure 4, *Ours* achieves comparable training efficiency to *ILAD*, even without the use of extra trajectory demonstrations, *Ours* can converge to a higher Success. *Ours* also surpasses other baselines of Success for both unseen and unseen category instances. This demonstrates the robust generalisation capabilities of our method. *Ours* also has the highest performance of Posture, which indicates that our method is better aware of the user's intentions. The qualitative results in Figure 3 b) demonstrate the successful grasping of objects across different user's intentions. As indicated in Table 1, *Ours* also causes the least disturbance to the object.

*ILAD* and *IBS* are considered as the strongest baseline. *ILAD* utilises additional trajectory demonstrations for RL training. However, *Ours* still surpasses this baseline because *ILAD* still relies on RL to generalise across diverse object categories. For *IBS*, we observed that *IBS* tends to fail when the human wrist trajectory has the potential to collide with the ground. This could be attributed to the reward design of *IBS*, which needs to balance between colliding with the scene and grasping the object. Additionally, computing the *IBS* representation is ten times more computationally expensive.

*PPO* primarily fails because it attempts to learn a general policy that can grasp most objects, as shown in Figure 3 a), which is not suitable for diverse objects and grasp poses. *PPO (Goal)* faces challenges due to its inability to adjust the goal during the approach phase. Since the wrist is continuously moving during the process, the goal set at the initial state may not be suitable.

|  | Seen Category | | Unseen Category | |
|---|---|---|---|---|
|  | Tran(cm) $\downarrow$ | Rot (rad) $\downarrow$ | Tran(cm) $\downarrow$ | Rot (rad) $\downarrow$ |
| PPO(Goal) | $2.621_{\pm 0.415}$ | $0.589_{\pm 0.038}$ | $2.537_{\pm 0.296}$ | $0.543_{\pm 0.040}$ |
| PPO | $2.745_{\pm 0.168}$ | $0.594_{\pm 0.045}$ | $2.771_{\pm 0.254}$ | $0.563_{\pm 0.039}$ |
| IBS | $2.653_{\pm 0.030}$ | $0.572_{\pm 0.002}$ | $2.596_{\pm 0.119}$ | $0.520_{\pm 0.011}$ |
| ILAD | $2.443_{\pm 0.042}$ | $0.548_{\pm 0.027}$ | $2.534_{\pm 0.101}$ | $0.515_{\pm 0.022}$ |
| Ours | $\mathbf{2.131}_{\pm 0.138}$ | $\mathbf{0.449}_{\pm 0.020}$ | $\mathbf{2.127}_{\pm 0.165}$ | $\mathbf{0.428}_{\pm 0.029}$ |

**Table 1:** Results of **Stability** on seen category unseen instances and unseen category instances. *Tran:* translation of objects from the initial position to the final position. *Rot:* rotation of objects from initial orientation to final orientation.

### 5.2 Ablation Studies and Analysis

We conduct the ablation studies to investigate: 1) *The effectiveness of decomposing the policy into primitive policy and residual policy.* 2) *The necessity of different action modules.* 3) *The impact of different action modules on final action.*

The ablation results depicted in Figure 5 demonstrate the significance of our proposed approaches. *Ours w/o GF* experiences a significant performance decline, indicating the significance of combining RL policy with primitive action. Similarly, *Ours w/o RL* shows a substantial drop in performance by focusing solely on the 'how' of grasping, which results in collisions during the grasping process.

Although *Ours w/o* $\mathbf{a}^r$ and *Ours w/o* $\mathbf{a}^s$ exhibit higher training efficiency, both methods eventually experience training collapse. During this collapse, we observe that the agent tends to either follow the primitive action more closely or deviate from it to a greater extent. This suggests that the combination of both action modules allows for a better policy learning process that effectively utilises the primitive action.

The results shown in Table 2 indicate that the primitive action, which does not consider collisions due to physical constraints in the grasping procedure, can achieve similar performance to *Ours*. This suggests that the primitive action module has a good understanding of how to grasp, but it moves quickly towards the target without considering potential collisions, as shown in Figure 6. The inclusion of the $\mathbf{a}^s$ module slows down the progress of $\mathbf{a}^p$ to avoid collisions, as illustrated in Appendix D.1. However, this conservative approach leads to a drop in success. Nevertheless, after further addition of $\mathbf{a}^r$, the performance increases to $56.50\%$, which is comparable to $\mathbf{a}^p$ w/o coll. By further combining the $\mathbf{a}^r$ module, the final action is corrected to achieve the 'how to grasp' knowledge learned by $\mathbf{a}^p$ while also incorporating correct 'when to grasp' knowledge.

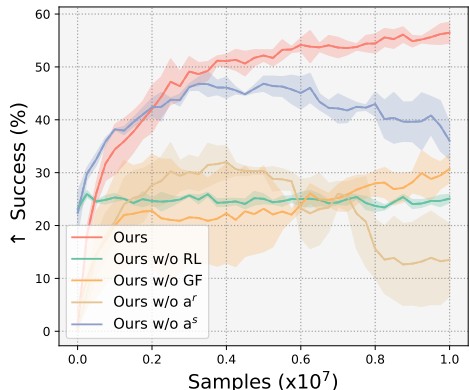

**Figure 5:** Ablation Study on decomposing policy and different action modules.

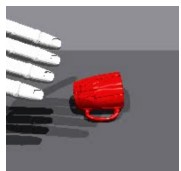 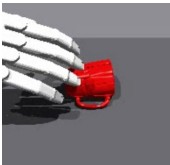 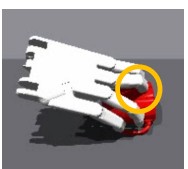 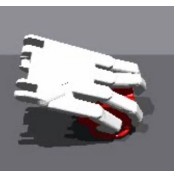

**Figure 6:** Qualitative results of grasping procedure governed by the primitive policy. The yellow circle highlights the collision between the finger and the object caused by premature closure.

| Action Type | Success |
|---|---|
| $\mathbf{a}^p$ | 19.49 % |
| $\mathbf{a}^p$ w/o coll | 55.60 % |
| $\mathbf{a}^p \odot \mathbf{a}^s$ | 5.08 % |
| $\mathbf{a}^p \odot \mathbf{a}^s + \mathbf{a}^r$ | 56.50 % |

**Table 2:** Success rate of different combinations of action modules.

## 5.3 Adaptability Results

To further demonstrate the adaptability of our method, we conduct both quantitative and qualitative experiments. Quantitatively, as shown in Table 3, we calculate the success rate for each object across five different grasp poses. For the objects that have been successfully grasped, *Ours* excels at grasping various parts of the objects. Qualitatively, Figure 7 demonstrates that the grasp pose generated by *Ours* closely resembles a human's intended grasp pose. Furthermore, we also show that GraspGF can adapt to wrist rotation; videos can be viewed at https://sites.google.com/view/graspgf.

| Success Rate | 1/5 | 2/5 | 3/5 | 4/5 | 5/5 |
|---|---|---|---|---|---|
| **Ours** | 13.67% | 27.70% | **31.41%** | **20.82%** | **6.40%** |
| **ILAD** | 25.44% | 32.53% | 26.61% | 12.53% | 2.89% |
| **IBS** | 27.70% | 34.73% | 25.67% | 10.29% | 1.61% |
| **PPO** | 48.20% | 35.49% | 13.10% | 3.01% | 0.20% |
| **PPO (Goal)** | 77.04% | 18.96% | 3.62% | 0.30% | 0.08% |

**Table 3:** Percentages of objects with varying success rates of grasping under different methods. The results are averaged over five different random seeds.

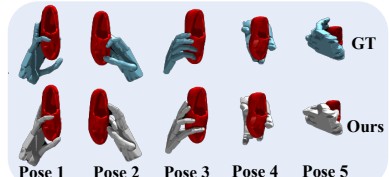

**Figure 7:** Qualitative results of grasping different parts of shoes with the same shape but different scales.

## 5.4 Real-world Results

In this section, we construct a real-world system to validate our method. As shown in Figure 8, the system consisted of calibrated multi-view RGB-D cameras (four Intel RealSense D415 sensors), a

UR10e robot arm equipped with a Shadow Hand as the end-effector, and an optical hand tracking sensor (Leap Motion Controller). To obtain the visual observation of the object, we reconstruct the scene point clouds from the multi-view RGB-D cameras. We only get the point cloud at the beginning of each grasping, assuming the point cloud observation does not change during the grasping. Then, we develop a teleoperation system inspired by the design of DexPilot [5] to enable the robot arm to track the trajectory of a human wrist, but with the difference of using a Leap Motion Controller as the hand tracking device to obtain human wrist and hand pose. It should be noted that we remove the estimated hand pose to emulate the human-assisting setting where only the wrist can be controlled by humans.

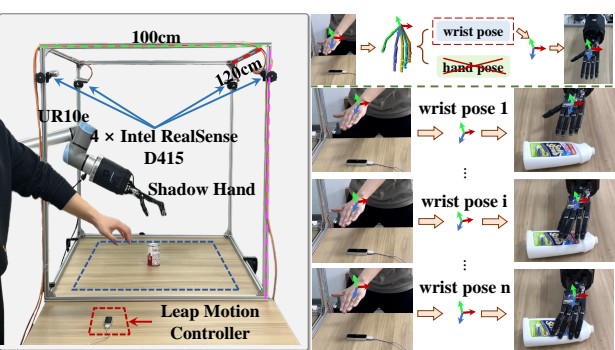

**Figure 8:** Real-world Experiment. Left: the real-world setup for real human-assisting dexterous grasping. Right: the qualitative result of real human-assisting dexterous grasping.

Following a similar setting in [16], we evaluated our policy in the real world using ten objects from the YCB dataset [31]. For each object, we selected two grasp poses that were intended to be achieved by a human. Each pose was tested five times, with a human randomly randomised initial object pose and wrist movement trajectories. The details of evaluation in the real world are deferred to D.2. As shown in Table 4, our policy achieved an average success rate of **66%** in real-world human-assisted grasping, demonstrating its generalisation ability in a real-world setting. The grasping policy tended to fail for certain large objects that were difficult for the dexterous hand to hold firmly, as well as for thin objects that were prone to colliding with the table. One of the reasons our policy still achieved a certain success rate in the real world is that when humans control the wrist of their hand, human will also try to adjust according to the grasping pose of the joint, which may help to grasp the object. Our real-world demonstrations can be viewed at https://sites.google.com/view/graspgf.

| Object | | | | | | | | | | | Average Success Rate |
|---|---|---|---|---|---|---|---|---|---|---|---|
| Pose 1 | 5/5 | 5/5 | 5/5 | 1/5 | 1/5 | 5/5 | 4/5 | 5/5 | 1/5 | 1/5 | **66%** |
| Pose 2 | 4/5 | 4/5 | 3/5 | 3/5 | 4/5 | 4/5 | 2/5 | 4/5 | 1/5 | 4/5 | |

**Table 4:** Results of success rate on 10 different YCB objects with two different poses in the real world.

## 6 Related Works

### 6.1 Dexterous Hand Grasping

Existing studies on dexterous hand robotics mainly focus on dexterous grasping, assuming full control over the hand, including both the hand base and joint movements by the agent [27, 32, 14, 29, 30].

However, Reinforcement Learning (RL) based methods struggle with generalisation to different objects due to object diversity, high-dimensional state and action spaces, and sparse rewards [33]. To address this issue, most existing approaches rely on human-collected demonstrations [17, 16], which require engineering-heavy retargeting. The most relevant work, IBS [27], combines RL with a human-designed representation that explicitly incorporates contact information between the hand and objects to aid in grasping. However, this computationally costly representation limits the practicality. Furthermore, in our proposed approach, we address a different setting where the movement of the hand base is determined by the user, requiring the agent to be 'user-centric' rather than 'object-centric'. Additionally, our approach only requires a set of grasp examples and is free of human design.

In the medical AI field, there are studies with a similar setting to ours, known as prosthetic dexterous grasping [11, 12, 13]. These studies aim to control a prosthetic hand to assist users in grasping and explore open-looped approaches that initially predict the grasp type and then execute predefined primitives for control. In contrast, we explore a closed-loop control policy that adaptively adjusts the grasp pose based on the current user-object relationship and the user's movement trajectory history.

## 6.2 Score-based Generative Models

In the pursuit of estimating the gradient of the log-likelihood associated with given data distribution, the score-based generative model, originally introduced by [34], has garnered substantial attention in the research community [22, 34, 35, 36, 37, 19, 38]. The denoising score-matching (DSM), as proposed by [22], further introduces a tractable surrogate objective for score-matching. To enhance the scalability of score-based generative models, [35] introduces a sliced score-matching objective that projects the scores onto random vectors before comparing them. Song et al. also introduce annealed training for denoising score matching [36], along with corresponding improved training techniques [37]. They also extend the discrete levels of annealed score matching to a continuous diffusion process and demonstrate promising results in image generation [19]. Recent works further explore the design choices of the diffusion process [39], maximum likelihood training [38], and deployment on the Riemann manifold [40]. These recent advances show promising results when applying score-based generative models in high-dimensional domains and promote wide applications in various fields, such as object rearrangement [21], medical imaging [41], point cloud generation [42], scene graph generation [43], point cloud denoising [44], depth completion [45], and human pose estimation [18]. These works formulate perception problems into conditional generative modelling or inpainting, allowing the utilisation of score-based generative models to address these tasks.

In contrast, our focus lies in the application of score-based generative models for training low-level control policies. In this domain, [46] and [20] have proposed learning diffusion models from offline trajectories and leveraging these models for model predictive control. However, these approaches suffer from the drawback of requiring a substantial amount of offline data and inefficiency during test-time sampling. To the best of our knowledge, our method represents the first exploration of score-based generative models for learning closed-loop dexterous grasping policies.

# 7 Conclusion

In this work, we introduce *human-assisting dexterous grasping*, wherein a policy is trained to assist users in grasping objects by controlling the robotic hand's fingers. To search for a *user-aware* policy, we propose a novel two-stage framework that decomposes the task into learning a primitive policy via score-matching and training a residual policy to complement the primitive policy via RL. In experiments, we introduce a human-assisting dexterous grasping environment that consists of 4900+ on-table objects with up to 200 realistic human wrist movement patterns. Results demonstrate that our proposed method significantly outperforms the baselines across various metrics. Our analysis reveals that our trained policy is more tailored to the user's intentions. Our real-world experiments indicate that our learned policy can generalise to the real world to some degree without fine-tuning.

**Limitations and Future works.** Our method takes the full point cloud as the visual observation, which is not accessible in the wild. In the future, we may leverage teacher-student learning to generalise our method to a partial observation setting.

**Ethics Statement.** Our method has the potential to develop home-assistant robots and assist individuals with hand disabilities, thus contributing to social welfare. We evaluate our method in simulated environments, which may introduce data bias. However, similar studies also have such general concerns. We do not see any possible major harm in our study.

## Acknowledgments and Disclosure of Funding

We would like to thank Qianxu Wang and Haoran Lu for supporting the real-world experiments. This work is supported by the National Natural Science Foundation of China - General Program (Project ID: 62376006), the National Youth Talent Support Program (Project ID: 8200800081) and the Beijing Municipal Science & Technology Commission (Project ID: Z221100003422004).

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

# A  Task Details

## A.1  Simulation

**Human Wrist Trajectory Generation**: We generate the human wrist trajectories using the following steps based on the human trajectory pattern dataset in A.2:

- Randomly select one object sample for each environment and add random noise to the x and y position change of the sampled object pose within the range of (-0.5, 0.5) meters.

- Sample a grasp pose that is different from the last sampled grasp pose for this object. We measure the distance between each grasp example of the object and the last sampled grasp example. If the sampled grasp example is already in the sample history, we randomly choose a new grasp example and clear the history.

- Based on the sampled grasp pose, we generate the initial wrist pose. First, we compute the centre of the fingertip pose and the wrist pose to obtain a vector pointing from the fingertip centre to the wrist. Then, we randomly sample the initial wrist position within a length range of 0.15 to 0.2 meters and randomly sample a deviation angle within the range of 0 to 20 degrees according to the vector.

- For the initial orientation, we randomly sample the delta angle change, denoted as $\delta_a$, from a distribution based on human rotation data. We set the initial angle as $A_{target}$ * (1 - $\delta_a$ / 2*$\pi$), where $A_{target}$ represents the target angle.

- After obtaining the initial and final poses of the wrist, we aim to add noise to the trajectories during training. However, simply adding random noise would result in a shaking wrist trajectory that does not resemble human behaviour. To address this, we randomly sample two normalized trajectory patterns $p1$ and $p2$ during training. We then combine these two patterns using a random coefficient $c$ sample from (0,1) to generate a new pattern: $c$ * $p1$ + (1 - $c$) * $p2$. This generated pattern is then used to define the trajectory of the wrist, which serves as the human wrist trajectory, as shown in Figure 9.

By following the above generation process, we can obtain human trajectories with diverse human-like grasping patterns, diverse velocities, and diverse initial hand poses for diverse objects, while guaranteeing that the agent can grasp the object at the final wrist pose.

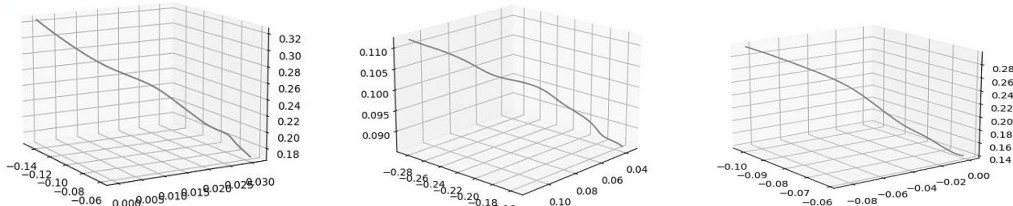

**Figure 9:** Visualizations of generated human trajectories.

**State Space**: For all joints $\mathbf{J}$ and under-actuated joints $\mathbf{J}^u$, we normalize the joint states to the range of (-1, 1) based on their respective joint limits. As for the wrist pose $\mathbf{b}$, we directly utilize the absolute pose relative to the world frame. Regarding the object point cloud $o$, we sample 1024 points from the object mesh.

**Action Space**: The action corresponds to the relative change applied to the 18 joints $\mathbf{J}$. Each action is clipped to the range of (-1, 1) and then scaled by a factor of 0.05.

## A.2  Datasets

**Grasp Example Dataset**: The ShadowHand model used in IBS is equipped with 18 degrees of freedom, including four under-actuated joints that are not directly controlled by the agent. However, in the table-top dataset presented in [10], it is assumed that the ShadowHand model has direct control over these four under-actuated joints. Consequently, the grasp examples from the dataset may not be directly applicable to our ShadowHand model. To address this, we incorporate the grasp poses from

the dataset into our simulation environment and reevaluate the grasp examples. We assess each grasp example based on the successful lifting of the object, with each grasp pose being tested twice for reliable evaluation.

To ensure a fair comparison, we select five of the most diverse grasp poses for each object from the re-filtered dataset for both seen category unseen instances and unseen category instances.

**Human Wrist Trajectory Pattern Dataset**: HandoverSim![28] provides handover trajectories collected from real humans, which can be utilized to generate human-like wrist trajectories during grasping in our task. However, these trajectories in HandoverSim![28] are only available for a limited set of objects and cannot be directly applied to all objects in our simulation environment. Therefore, to generate human-like grasping wrist trajectories in our simulation environment, we first collect wrist movement trajectories from the initial pose to the pose where the object is successfully grasped. To capture the movement patterns, we normalize the trajectories based on the max and min values of each axis (x,y,z). Additionally, for the rotation component, we collect the distribution of wrist rotation change from the initial orientation to the final orientation of each axis (roll, pitch, yaw). We generate human trajectories according to these patterns.

### A.3 Evaluations

For baseline comparison and ablation study, we conducted evaluations using 5 random seeds. For each seed, the evaluation of training instances was performed five times. The grasp pose for each object is different in each iteration. Throughout the evaluations, we utilized the human wrist trajectory patterns randomly sampled from their respective sets without introducing any additional noise. The human trajectory remained consistent across all comparisons for the same random seed.

## B  Implementation Details

### B.1  Primitive Policy

**Training Objective:** In practice, we adopt an extension of DSM [19] that estimates a *time-dependent score network* $\mathbf{s}_\omega(\mathbf{J}|o^*, \mathbf{b}^*, t)$ to denoise the perturbed data from different noise levels simultaneously:

$$\mathcal{L}(\omega) = \mathbb{E}_{t \sim \mathcal{U}(\epsilon, T)} \left\{ \mathbb{E}_{\substack{\widetilde{\mathbf{J}} \sim q_{\sigma(t)}(\widetilde{\mathbf{J}}|\mathbf{J}), \\ (\mathbf{J}^*, o^*, \mathbf{b}^*) \sim \mathcal{D}_{\text{success}}}} \lambda(t) \left[ \left\| \mathbf{s}_\omega(\widetilde{\mathbf{J}}, |o^*, \mathbf{b}^*, t) - \frac{1}{\sigma^2(t)}(\mathbf{J} - \widetilde{\mathbf{J}}) \right\|_2^2 \right] \right\} \quad (8)$$

where $T = 1$, $\epsilon = 10^{-5}$, $\lambda(t) = \sigma^2(t)$, $\sigma(t) = 1 - e^{-\frac{1}{2}t^2(\beta_{max} - \beta_{min}) - t\beta_{min}}$ and $\beta_{max} = 10, \beta_{min} = 0.1$ are hyper-parameters. The optimal time-dependent score network holds $\mathbf{s}_\omega^*(\mathbf{J}|o^*, \mathbf{b}^*, t) = \nabla_{\mathbf{x}} \log q_{\sigma(t)}(\mathbf{J}|o^*, \mathbf{b}^*)$ where $q_{\sigma(t)}(\mathbf{J}|o^*, \mathbf{b}^*)$ is the perturbed data distribution:

$$q_{\sigma(t)}(\mathbf{J}|o^*, \mathbf{b}^*) = \int q_{\sigma(t)}(\widetilde{\mathbf{J}}|\mathbf{J}) p_{\text{success}}(\mathbf{J}|o^*, \mathbf{b}^*) d\mathbf{J} \quad (9)$$

With the trained score network $\mathbf{s}_\omega$, we parameterize the primitive policy as:

$$\pi_p^\theta(\cdot|\mathbf{J}, o^*, \mathbf{b}^*) = \mathbf{s}_\omega(\mathbf{J}|o^*, \mathbf{b}^*, 0.005) \quad (10)$$

**Network Architecture:**

We utilize the Gradient Field backbone introduced in [21] along with the PoinetNet++ backbone from [47] to build our chosen backbone for this study.

For training the primitive policy, we initially extracted a maximum of five distinct poses for each object within the training instances to balance the training data, resulting in a total of 15,387 grasping examples. It is important to note that we exclusively employ grasp examples for the training of the primitive policy, without incorporating any trajectories. We use the Adam optimizer with a learning rate of 2e-4 for training, The batch size is set to the total number of grasping examples divided by 5. It takes 60 hours to train on a single A100 for primitive policy to converge.

## B.2 Residual Policy

To incorporate object geometry information into the residual policy, we employ PointNet[25]. We begin by pretraining PointNet[25] using the same procedure as the primitive policy training with PointNet++[24]. Subsequently, during training for the residual policy, we continue fine-tuning PointNet[25].

**Reward Function**: Based on empirical observations, we found that the following intrinsic reward function (Eq. 11) which encourages the agent to simply follow the direction of primitive policy will lead to faster convergence.

$$r_{\text{sim}} = \lambda_a \cdot \left\langle \frac{\mathbf{a}^p}{|\mathbf{a}^p|}, \mathbf{J}_t - \mathbf{J}_{t-1} \right\rangle \tag{11}$$

Note that since the intrinsic reward $r_{\text{sim}}$ is the only dense reward in our method, we set its frequency to 5 to avoid overfitting to the primitive policy. Empirically, we set $\lambda_s = 1.0$, $\lambda_a = 0.09$, and $\lambda_h = 0.5$ for our approach.

**RL Backbone**: We implemented the Proximal Policy Optimization (PPO) algorithm [23] using the PyTorch implementation provided in [2]. In our implementation, we incorporated the PointNet backbone from [17] into the PPO backbone.

**Hyper-parameters:** We mainly kept the hyper-parameters of PPO the same as those in [2], with the following exceptions: we set the number of update intervals ($nsteps$) to 50, the number of optimization epochs ($noptepochs$) to 2, the mini-batch size mini_batch_size($mini\_batch\_size$) to 64, and the discount factor ($gamma$) to 0.99.

We trained the residual policy for a total of 10 million agent steps, which took approximately 15 hours using a single A100 GPU.

## C   Baselines Implementations

**IBS**: In the IBS baseline [27], two buffers are maintained: a demonstration buffer for warm-up RL training and an experiment buffer for storing the transition tuples. We set the size of both buffers to 15,000, which is three times larger than in our method. To generate the demonstration buffer, we follow a similar procedure in IBS [27]. Specifically, we randomly sample grasp poses from the same data used to train the primitive policy. We then use linear interpolation to compute joint actions based on the initial and target joint states. By applying these actions with the generated human trajectories, we store the transition tuples in the demonstration buffer until it reaches the maximum size.

**PPO**: Since the PPO baseline does not have a primitive policy, we substitute the intrinsic reward with the fingertip distance reward. It is important to note that we keep the $r_h$ and success reward the same as in our approach. We use PointNet++ [24] for this baseline, as the PointNet may lead to training collapse. All other training parameters remain the same as in our approach.

**PPO(Goal)**: Similar to the PPO baseline, we use PointNet++ [24] and keep all training parameters the same as in our approach. We introduce a goal pose matching reward that computes the distance between the current joint position and the goal joint position, in addition to the rewards used in PPO.

**ILAD**: In the ILAD baseline [17], additional trajectories are required to train RL. We use the same data used to train the primitive policy to generate these trajectories following the same process as described in ILAD [17]. In total, we generate 125k transition tuples. We use PointNet [25] for this baseline and keep all training parameters the same as in our approach. The reward remains the same as PPO.

## D   Additional Results

### D.1   Additional Ablations

**The necessity of $\mathbf{a}^s$ module:**

As shown in Figure 10, we visualise the grasp pose of the final state using different action modules $\mathbf{a}^p$ and $\mathbf{a}^p \odot \mathbf{a}^s$, the results are generated without simulating the collision between hand and object in the simulation environment. We can observe that $\mathbf{a}^p$ will usually cause a collision between the hand

and the object. When using the combination of $\mathbf{a}^s$, the finger still moves towards the similar grasp pose as with $\mathbf{a}^p$, while the $\mathbf{a}^s$ module will also try to avoid the collision with the object by slowing down the movement of the $\mathbf{a}^p$.

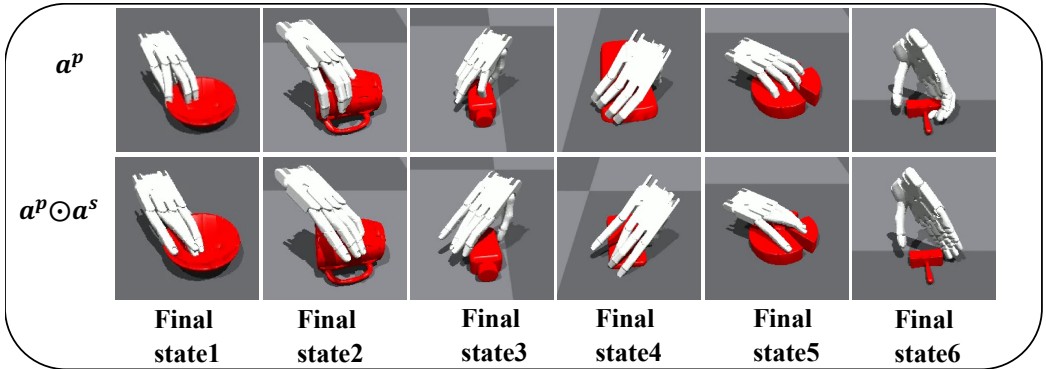

**Figure 10:** Qualitative result of final grasp pose using $\mathbf{a}^p$ and $\mathbf{a}^p \odot \mathbf{a}^s$ action module

**The effectiveness of intrinsic reward $r_{\mathbf{sim}}$:**

As the primitive policy provides good guidance on "how to grasp", we introduce the $r_{sim}$ reward to encourage the final output action $a^p \odot a^s + a^r$ to explore in direction of the primitive policy's action $a^p$, this reward is the inner product of $\frac{a^p}{\|a^p\|_2}$ and $J_t$ - $J_{t-1}$, where $J$ signifies the state of finger joints. A higher value of $r_{sim}$ indicates a smaller angle between the gradient and $J_t$ - $J_{t-1}$, implying a greater similarity in the movement of finger joints to the gradient.

As shown in Figure 11, when the intrinsic reward $r_{\text{sim}}$ is not included, the success rate of our method drops, and the training efficiency is significantly lower compared to when $r_{\text{sim}}$ is included. This indicates that $r_{\text{sim}}$ plays a crucial role in guiding the agent to initially follow the primitive policy and helps accelerate the learning of the residual policy. Furthermore, the inclusion of $r_{\text{sim}}$ encourages the residual policy to follow the primitive policy throughout the entire training process, leading to better utilisation of the generalisation capabilities of the primitive policy.

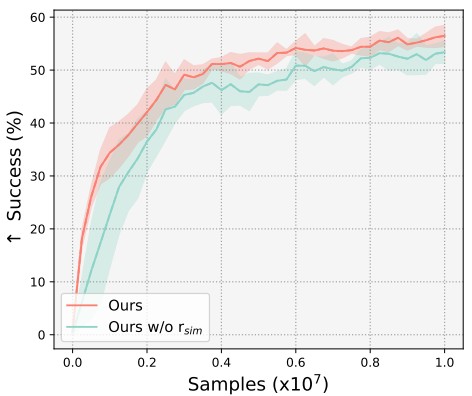

**Figure 11:** Ablation Study on intrinsic reward $r_{\text{sim}}$

**Action of each policy during the grasping process:**

The primitive policy's actions solely depend on object-wrist relative orientation. Thus when the hand is far away, the primitive policy will still close the fingers, as shown in Figure 12 (a) Stage1. As the hand's posture progressively approaches the target grasp pose, the mean value of the primitive policy's actions decreases, as shown in Figure 12 (a) Stage2.

To further understand the action of residual policy, we utilise a measure called $r_{sim}$. A higher value suggests the final action more closely follows the primitive policy. As shown in Figure 12 (b) Stage1, when the hand is far from the object, the residual policy will restrict the finger's early closure to prevent the collision, as the hand approaches the object, residual policy start to follow primitive policy, as shown in Figure 12 (b) Stage2. However, as shown in Figure 12 (b) Stage3, as the hand is about to grasp the object, the $r_{sim}$ starts to decrease. At the last few steps, the residual policy will further refine the pose to hold the object firmly, leading to the negative $r_{sim}$ value.

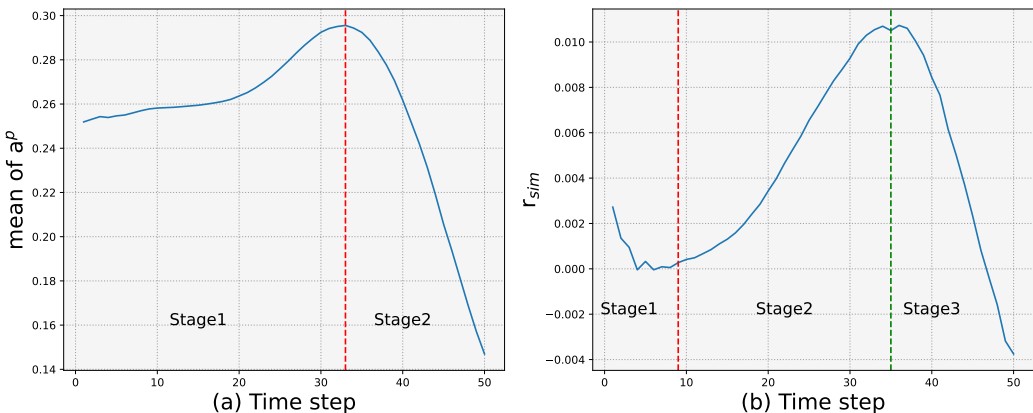

**Figure 12:** (a) Variation of the mean value of the primitive policy's action over time steps. (b) Variation of the $r_{sim}$ reward over time steps.

## D.2    Real World Evaluation

During the real-world grasping procedure, once the human wrist was detected to lift, the hand would automatically lift up, move towards a target box, and open the fingers to release the object, as illustrated in Figure 13. A grasp was considered successful if the object was lifted and moved to the box without falling before the fingers opened.

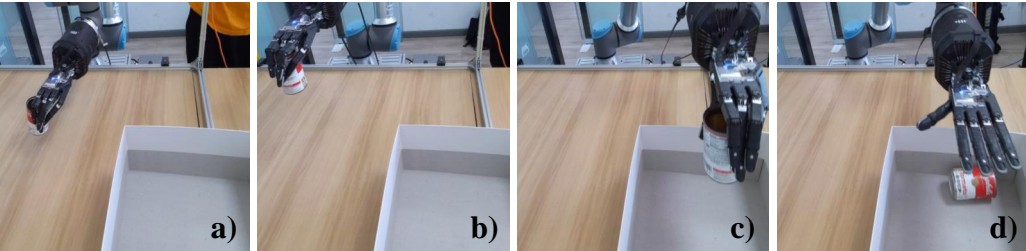

**Figure 13:** Real-World evaluation process for assessing grasp pose success: a) Detection of human wrist lift. b) Hand automatically lift up. c) Hand automatically move to the target box. d) Hand open to release the object.

## D.3    Inference Speed for Each Module

We evaluate the inference speed on the GTX 1650, which is also used in our real-world experiment. We set the batch size equal to 1 and ran the policy 50 times to obtain a reliable average time for the inference speed of each module. As shown in Table 5, both modules take less than 0.004 seconds for each inference. This indicates that our integrated system is capable of seamless human interaction.

| Device | Inference time for primitive policy | Inference time for residual policy |
|---|---|---|
| **GTX1650** | 0.002766s | 0.003646s |

**Table 5:** Results of inference speed of primitive policy and residual policy on GTX1650.

## D.4    Robustness to Perception Noise

To demonstrate the robustness of our method, we inject two levels of noise to the wrist pose observation following [48]. As indicated in Table 6, our approach yields comparable outcomes

with a 2-degree/2-cm estimation error while exhibiting approximately 10% reduction in performance under a 5-degree/5-cm error threshold, which indicates that our method can handle estimation error to some degree.

| standard deviation of noise | $0°$, 0cm | $2°$, 2cm | $5°$, 5cm |
|---|---|---|---|
| success rate | 56.69% | 55.24% | 44.35% |

**Table 6:** Results of success rate under different levels of observation noise on unseen category instances using seed 0. The Gaussian noise is determined by the standard deviation of the specified threshold and added to observations of wrist position and orientation separately, any noise exceeding these bounds will be clipped. The pose estimation method can achieve 80.99% accuracy under $2°$, 2cm, and achieve 95.80% accuracy under $5°$, 5cm.

