# OpenReview forum: "Learning Score-based Grasping Primitive for Human-assisting Dexterous Grasping"
_NeurIPS.cc/2023/Conference — NeurIPS 2023 poster_

### Official Review · Reviewer_GtH3 · 2023-06-25

**Soundness:** 4 excellent
**Presentation:** 4 excellent
**Contribution:** 4 excellent
**Rating:** 7
**Confidence:** 3

**Summary:**

This paper introduced a novel and challenging task that performs dexterous grasping according to human wrist movements. This task is potentially useful for applications with prosthetic hands.

The paper further proposed a novel two-stage framework that solves the two challenging aspects of the proposed task and demonstrated strong performance in both simulated and real-world experiments.

**Strengths:**

The proposed task is novel and potentially helpful to social welfare. The proposed framework is intuitive and is properly designed for the challenges of its task. The authors have also conducted extensive experiments to show the capacities of the proposed method.

The paper is well-structured and written.

**Weaknesses:**

From the qualitative results in the supplementary video, I noticed that for most objects, the graspings are from the same angle relative to the object. For example, with the chips can, all demonstrated graspings are from the side of the cylinder regardless of how the can is placed. This makes me wonder if the proposed method can truly adapt to different approach angles and different *user intentions*. Please correct me if I missed anything from the videos.

**Questions:**

1. For the baseline methods, are all baselines re-trained to take the wrist pose as a condition? I didn’t find this piece of information in the paper.
2. In figure 5, why do w/o $a^r$ and w/o $a^s$ success rates drop as more samples are seen after 5e6 samples?

**Limitations:**

1. The method assumes full point cloud observation which may limit its application in the real world.
2. The qualitative results did not show how the proposed method adapts to different wrist poses relative to the object.

---

> ### Author Rebuttal · Authors · 2023-08-08
>
> > **Q1: From the qualitative results in the supplementary video, I noticed that for most objects, the graspings are from the same angle relative to the object. For example, with the chips can, all demonstrated graspings are from the side of the cylinder regardless of how the can is placed. This makes me wonder if the proposed method can truly adapt to different approach angles and different user intentions. Please correct me if I missed anything from the videos.**：
>
> Thank you for bringing up this point. While the approach angles appear similar in the videos, it's important to note that the object's relative pose to the wrist is actually varied due to changing object poses. Principally, we have a world coordinate, and a wrist coordinate. The graspings looks like same angle relative to the object is because the movement of hand is similar in world coordinate. However, since the object pose in world coordinate is also changing (for instance, the Bleach Cleanser being positioned both vertically and horizontally), the object's pose in the wrist coordinate is consequently changing, resulting in different relative poses. Regarding the Chips Can, despite all demonstrations involving side grasps, it's essential to recognize that the object's relative pose to the wrist is in fact distinct.
>
> To further illustrate the adaptability of our method to various approach angles and user intentions, we have conducted two additional experiments. As detalied in **Section 1 of the anonymous project page**, participants first move the hand to either the middle or the right side of the object, then start to approach. When dealing with chips, participants attempt to grasp the top of the chip, while for the mug, participants attempt to grasp the mug's edge. Alongside these real-world experiments, we have also conducted more comprehensive simulation experiments, as detailed in the [Q2 of Common Response](https://openreview.net/forum?id=fwvfxDbUFw&noteId=DtPbwqFxR8), to effectively demonstrate the adaptability of our method.
>
> > **Q2: For the baseline methods, are all baselines re-trained to take the wrist pose as a condition? I didn’t find this piece of information in the paper.**:
>
> Thank you for pointing this out. To clarify, all baselines are re-trained by taking latest 5 frame wrist poses as input under human-assisting setting (e.g., the agent can not move the wrist), which aligns with "Ours" experiment setup. We will add additional detail of  this in the future revision.
>
> > **Q3: In figure 5, why w/o $a^r$ and w/o $a^s$ success rates drop as more samples are seen after 5e6 samples?**
>
> Thank you for raising this concern. We hypothesis that this issue is tied to the fundamental exploration and exploitation challenge in classical reinforcement learning. As the primitive policy provides a strong guidance for the reinforcement learning algorithm, the agent might tend to focus solely on exploitation based on the primitive policy. This behavior can lead to getting stuck in local minima and ultimately result in a decrease in the success rate. Nevertheless, with the incorporation of both a_r and a_s, our policy is better equipped to strike a balance between exploration and exploitation. This balanced approach allows the agent to navigate both the exploration of novel strategies and the exploitation of existing knowledge more effectively, leading to improved overall performance.
> > **Q4: The method assumes full point cloud observation which may limit its application in the real world.**:
>
> Thank you  for bring this up. Due to the page limit, please refer to [Q4 of Common Response](https://openreview.net/forum?id=fwvfxDbUFw&noteId=DtPbwqFxR8).
>
> > **Q5: The qualitative results did not show how the proposed method adapts to different wrist poses relative to the object.**:
>
> Thank you  for bring this up. We have conducted additional experiments to demonstrate the adaptability of our method. Due to the page limit, please refer to [Q2 of Common Response](https://openreview.net/forum?id=fwvfxDbUFw&noteId=DtPbwqFxR8).

---

### Official Review · Reviewer_PG5Z · 2023-07-06

**Soundness:** 3 good
**Presentation:** 3 good
**Contribution:** 3 good
**Rating:** 6
**Confidence:** 4

**Summary:**

This paper introduces a novel task called human-assisting dexterous grasping, which aims to train a policy for controlling a robotic hand's fingers to assist users in grasping objects. Unlike conventional dexterous grasping, this task is more complex as the policy must adapt to diverse user intentions and the object's geometry. The proposed approach consists of two sub-modules: Grasping Gradient Field (GraspGF) and a history-conditional residual policy. GraspGF learns 'how' to grasp by estimating the gradient of a synthesized success grasping example set, while the residual policy determines 'when' and at what speed the grasping action should be executed based on the trajectory history. Experimental results show that the proposed method outperforms baselines in terms of user-awareness and practicality in real-world applications.

**Strengths:**

This paper's strengths can be outlined as follows:

1. Introduction of a unique dexterous grasp task involving shared autonomy between humans and robots, a topic not extensively explored in prior research.
2. Application of the Denoising Score Matching method to the grasping task.
3. Explicit representation of robot finger velocity.
5. A thorough acknowledgment of the system's limitations, including the requirement for a complete point cloud.


**Weaknesses:**

However, the paper also has some drawbacks:

1. The proposed method is better suited for teleoperation settings compared to the reinforcement learning (RL) baselines used in the experiments. It is essential to include comparisons to teleoperation methods without assisted grasping, both qualitatively and quantitatively.
2. The paper's presentation could be enhanced. For instance, the individual images in Figure 2 could be better explained, as it is currently difficult to comprehend and not highly informative.
3. The residual policy, which corrects the primitive policy's action, does not consider the primitive policy action as input. This seems illogical for predicting velocity and bias terms without knowing the direction.


**Questions:**

Regarding $r_{sim}$ in Equation 7, more clarification on its functionality would be helpful. Additionally, it would be beneficial to know the inference speed for each module. For human-assistance, quick response times are crucial for seamless human interaction. Since visual modules are utilized, a profiling analysis may be necessary. I am happy to raise the score if the concerned are addressed adequately.

## After Rebuttal
------------------------
The author response looks great to me. Some of the presentation issues are also addressed during the rebuttal phase. I agree with the author that this is more focus on assisting upper limb amputees with prosthetic hands instead of assisting normal persons. I would like to raise the score and glad to see it is accepted.

**Limitations:**

To address the weaknesses, the paper's authors could improve the presentation by creating more self-contained figures. Furthermore, the hand/object in several images, such as Figure 3, is too small to clearly discern the interaction patterns.

---

> ### Author Rebuttal · Authors · 2023-08-08
>
> > **Q1: The proposed method is better suited for teleoperation settings compared to the reinforcement learning (RL) baselines used in the experiments. It is essential to include comparisons to teleoperation methods without assisted grasping, both qualitatively and quantitatively.**:
>
> Apologies for the confusion. To clarify, the main motivation and practical application of assisting grasping is assisting upper limb amputees with prosthetic hands instead of assisting normal persons, as show in **Figure 1 of main pdf**. Traditional teleoperation method[1] is unsuitable for assisting upper limb amputees to grasp, because we can not get information of human finger. In real-world experiments, we use teleoperation solely to get human wrist poses for mimicking grasping by upper limb amputees. We will elaborate the difference of teleoperation and human-assisting grasping to the introduction section of main pdf in the future revision.
>
> > **Q2:  The paper's presentation could be enhanced. For instance, the individual images in Figure 2 could be better explained, as it is currently difficult to comprehend and not highly informative.**:
>
> Apologies for the confusion. We will add additional explanations and details to the pipeline shown in Figure 2 of the main PDF. This will include more detailed annotations that explicitly indicate how the primitive policy's actions are utilized as input for the residual policy. We will also provide further illustration of the functionality of the two components, as well as more compact and clear element in each image. Thanks again for your valuable suggestion!
>
> > **Q3: The residual policy, which corrects the primitive policy's action, does not consider the primitive policy action as input. This seems illogical for predicting velocity and bias terms without knowing the direction.**:
>
> Thanks for reminding this. We would like to clarify that the residual policy does take the primitive policy's action as an input. The process involves the primitive policy initially taking the joint state $J_t$, object point cloud $o_t$, and wrist pose $b_t$ as inputs to generate the primitive action $a^p$. Subsequently, the residual policy takes both the primitive action $a^p$, joint state $J_t$, object point cloud $o_t$, and wrist trajectories $H_t$ as inputs to produce the residual actions $a^s$ and $a^r$.
> We are sorry for the confusion by simplification of notation in the initial presentation. In the revised version of our paper, we will provide a more detailed and accurate description of the relationship between the primitive policy's action and the residual policy. Additionally, in the revised version of Figure 2, we will include detailed annotations that explicitly indicate how the primitive policy's action is utilized as input for the residual policy.
>
> > **Q4: Regarding $r_{sim}$ in Equation 7, more clarification on its functionality would be helpful**:
>
> Thank you for pointing this out. As the primitive policy provides good guidance on "how to grasp," we introduce the $r_{sim}$ reward to encourage the final output action ${a^p} \odot {a^s} + {a^r}$ to explore in direction of the primitive policy's action $a^p$ , this reward is the inner product of $\frac {a^p}{{\parallel {a^p} \parallel}2}$ and $J_t$ - $J_{t-1}$, where $J$ signifies the state of finger joints. A higher value of $r_{sim}$ indicates a smaller angle between the gradient and $J_t$ - $J_{t-1}$, implying a greater similarity in the movement of finger joints to the gradient. Actually, we have already conducted an ablation study of $r_{sim}$ in **Section 4.1 of the Supplementary** to demonstrate how $r_{sim}$ facilitates the acceleration of the residual policy's learning process, and leads to a more effective utilization of the generalization capabilities of the primitive policy.
>
> > **Q5: Additionally, it would be beneficial to know the inference speed for each module. For human-assistance, quick response times are crucial for seamless human interaction. Since visual modules are utilized, a profiling analysis may be necessary.**:
>
> Thanks for bring up this concern. We evaluate the inference speed on the GTX 1650, which is also used in our real world experiment.
> We set the batch_size equal to 1 and ran the policy 50 times to obtain a reliable average time for the inference speed of each module.
> As shown in **Table 2 of the rebuttal pdf**, both modules take less than 0.004 seconds for each inference.
> This indicates that our integrated system is capable of seamless human interaction.
>
> **Reference**:
>
> [1] Handa, Ankur, et al. "Dexpilot: Vision-based teleoperation of a dexterous robotic hand-arm system."

---

> > ### Comment · Reviewer_PG5Z · 2023-08-11
> > **After Rebuttal Reviewer Response**
> >
> > The author response looks great to me. Some of the presentation issues are also addressed during the rebuttal phase. I agree with the author that this is more focus on assisting upper limb amputees with prosthetic hands instead of assisting normal persons. I would like to raise the score and glad to see it is accepted.

---

> > > ### Author Response · Authors · 2023-08-18
> > > **Thank You!**
> > >
> > > Thanks for raising your rating to 6. We are so glad that our responses help address your concerns. Thanks again for all your valuable feedback!

---

### Official Review · Reviewer_KjWd · 2023-07-06

**Soundness:** 3 good
**Presentation:** 2 fair
**Contribution:** 3 good
**Rating:** 5
**Confidence:** 4

**Summary:**

This paper focuses on addressing a task called human-assisting dexterous grasping. The aim is to create a finger controller to grasp objects with the robot's wrist conditioned on a human user's wrist. The authors propose 1) a Grasping Gradient Field (GraspGF) which estimates the gradient of a synthetic grasping example, and 2) a residual policy achieved through reinforcement learning. Experimental results demonstrate the superiority of the proposed method over previous ones.


**Strengths:**

The authors are tackling an interesting problem - guiding a robot hand to follow human wrist trajectories and utilizing a learnt finger controller to manipulate objects. This bears resemblance to teleoperation but only provides wrist information. I would appreciate further discussion on this aspect.

The authors introduce a score-matching-based method for learning a primitive policy and a residual policy to aid the primitive policy. This combines synthetic data with reinforcement learning to accelerate training and achieve better performance.

Authors have conducted a large number of real-world robotic experiments, showcasing the practical applicability of the proposed method.

**Weaknesses:**

While I agree that human-assisting dexterous manipulation holds potential, it is concerning if this work only addresses the grasping task without considering other dexterous manipulation problems. What are the specific differences and motivations between an automatic dexterous grasping method and user-provided wrist? What is the practical application? If, as the authors suggest, grasping different parts meets varying needs, could the authors conduct experiments to demonstrate this? Or, stepping back, could the proposed method grasp the part that the user intends to grasp? Would it be possible to conduct experiments on it?

In Table 2, 'ap w/o coll' seems to achieve similar performance, and considering the increment from 55.6% to 56.5%, the residual policy seems not necessary.

The authors should continue to polish the paper. For instance, the subscript 't' in 'a' on lines 161, 163, and 165 lacks consistency. The formatting of Table 2 could also be improved.



**Questions:**

What are the outputs of GraspGF when the hand is at different stages, such as when the hand is far from the object at t_0 and close to the object at t_n? What actions are produced in these instances?

Given the same initial state and wrist trajectory, can we achieve diversified results?

Observing Figure 4 and Table 1, there is not much difference between the seen and unseen conditions. Could the authors attempt to analyze this?

Other datasets, such as DexYCB, could provide human wrist trajectory and thus increase the current 200 trajectories.

**Limitations:**

What is the tolerance for errors in wrist estimation? Often, people cannot carefully move their wrists or do not have a precise estimation tool like Leap Motion.
Further, Leap Motion has a requirement for a complete hand without occlusion, which make me doubt about the algorithm's ability to help people with hand disabilities. It might be beneficial to consider alternative wearable sensors for wrist SE3 estimation or additional vision algorithm for wrist pose estimation based on RGB input.

---

> ### Author Rebuttal · Authors · 2023-08-08
>
> > **Q1: This bears resemblance to teleoperation ...? ..differences and motivations between an automatic dexterous grasping method...? What is the practical application?**:
>
> Thank you  for bring this up. Due to the page limit, please refer to [Q1 of Common Response](https://openreview.net/forum?id=fwvfxDbUFw&noteId=DtPbwqFxR8).
>
> > **Q2: ... it is concerning if this work only addresses the grasping task without considering other dexterous manipulation problems**:
>
> Thank you for raising this concern. Grasping serves as the foundational skill in manipulation tasks[1] ,such as picking up the hammer is the first step in nailing. Developing a generalized grasping algorithm forms the basis for more intricate manipulation tasks.
>
> In fact, our current framework can also be adapted for manipulation. For example, we can collect expert demonstrations for manipulation tasks and employ diffuser[2] for learning primitive policies. Then combines with the residual policy for further refinement during the manipulation process.
>
> > **Q3: If, as the authors suggest, grasping different parts meets varying needs...could the proposed method grasp the part that the user intends to grasp? ...**:
>
> Thank you  for bring this up. We have conduct additional experiments to demonstrate the adaptability of our method. Due to the page limit, please refer to [Q2 of Common Response](https://openreview.net/forum?id=fwvfxDbUFw&noteId=DtPbwqFxR8).
>
> > **Q4: In Table 2, 'ap w/o coll' seems to achieve similar performance...**:
>
> Thank you  for bring this up. We would like to clarify that residual policy is necessary. Due to the page limit, please refer to [Q3 of Common Response](https://openreview.net/forum?id=fwvfxDbUFw&noteId=DtPbwqFxR8).
>
> > **Q5: The authors should continue to polish the paper...**:
>
> Thank you for your suggestion.  We will review the formatting of the main pdf and ensure the consistency of the notation in the future revision.
>
> > **Q6:  What are the outputs of GraspGF when the hand is at different stages...**:
>
> The primitive policy's actions are solely depend on object-wrist relative orientation. Thus when the hand is far away, the primitive policy will still close the fingers, as shown in **Figure 3 (a)  Stage1 of the rebuttal pdf**.
> As the hand's posture progressively approaches the target grasp pose, the mean value of the primitive policy's actions decrease, as shown in **Figure 3 (a)  Stage2 of the rebuttal pdf**.
>
> To further understand action of residual policy, we utilize a measure called "$r_{sim}$".  Higher $r_{sim}$ value suggests the final action more closely follows the primitive policy. As shown in **Figure 3 (b) Stage1 of the rebuttal pdf**, when the hand is far from the object, the residual policy will restrict finger's easrly closure to prevent collision, as the hand approach the object, residual policy start to follow primitive policy, as shown in **Figure 3 (b) Stage2 of the rebuttal pdf**. However, as shown in **Figure 3 (b) Stage3 of the rebuttal pdf**, as the hand is about to grasp the object, the "$r_{sim}$" start to decrease. At the last few steps, reisudal policy will further refine the pose to hold the object firmly, leading to the negative "$r_{sim}$" value.
>
> > **Q7: Given the same initial state and wrist trajectory, can we achieve diversified results?**:
>
> We would like to clarify that our module is deterministic, adhering to the standard implementation of PPO.
>
> > **Q8: Observing Figure 4 and Table 1, there is not much difference between the seen and unseen conditions...**:
>
> Filtered from unidexgrasp[3], the current grasp dataset involves 3000+ training objects, totaling 0.36 million grasp poses. We hypothesize that this comprehensive dataset sufficiently captures diverse data distributions. Consequently, the distribution of seen and unseen datasets might be i.i.d with the training dataset, results in low difference. This observation aligns with our findings from unidexgrasp.
>
> > **Q9: Other datasets, such as DexYCB, could provide human wrist trajectory...**:
>
> Thanks for your valuable suggestion. We actually use DexYCB as human trajectory dataset and extract trajectories from HandOverSim by default configurations. To increase the diversity, we have augmented the dataset by fusing pairs of trajectories  (See **Section 1.1 of Supplementary**). Also, we could extract more human wrist trajectoies by changing the configurations of HandOverSim, as suggested.
>
> > **Q10: What is the tolerance for errors in wrist estimation? ...**:
>
> Thank you for bringing up this concern. To demonstrate the robustness of our method, we inject two levels of noise to the wrist pose observation following [4].
> As indicated in **Table 3 in rebuttal pdf**, our approach yields comparable outcomes with a 2-degree/2-cm estimation error, while exhibiting approximately 10% reduction in performance under a 5-degree/5-cm error threshold, which indicate that our method can handle estimation error to some degree.
>
> > **Q11:  Further, Leap Motion has a requirement for a complete hand without occlusion...**:
>
> Thanks for your valuable suggestion! We agree that occlusion is a major issue when deploying our system in the real world and it would be a great future direction. Actually, we are developing a system involves mounting a camera on the user's head and estimating the wrist pose using an instance-level pose estimation method.
>
> **Reference**:
>
> [1] Newbury, Rhys, et al. "Deep learning approaches to grasp synthesis: A review."
>
> [2] Janner, Michael, et al. "Planning with Diffusion for Flexible Behavior Synthesis."
>
> [3] Xu, Yinzhen, et al. "Unidexgrasp: Universal robotic dexterous grasping via learning diverse proposal generation and goal-conditioned policy."
>
> [4]Chen, Hansheng, et al. "Epro-pnp: Generalized end-to-end probabilistic perspective-n-points for monocular object pose estimation."

---

> > ### Comment · Reviewer_KjWd · 2023-08-18
> >
> > Thank you for the author's rebuttal. After reading the reviews from other reviewers and the author's responses, I maintain my borderline accept rating.

---

> > > ### Author Response · Authors · 2023-08-19
> > >
> > > Thank you for your response. If you have any additional concerns, we are glad to address them.

---

### Official Review · Reviewer_ju1F · 2023-07-07

**Soundness:** 3 good
**Presentation:** 3 good
**Contribution:** 3 good
**Rating:** 6
**Confidence:** 5

**Summary:**

This paper proposes a new task called assisting grasping. The main difference between this task and classical dexterous grasping is the wrist movement is controlled by a human instead of by the grasping algorithm. The authors propose a two stage method to solve this problem. First, they learn the grasping skill using a successful grasping dataset via score-matching loss. Then, they fine-tune this policy using RL in simulation. They show the proposed method is better than pure RL and score-matching is useful compared to imitation learning algorithms.

**Strengths:**

(+) The authors formulate the learning from a set of successful grasps as a denoising problem, which is quite interesting and novel. I think this is an effective design choice.

(+) This paper proposes to separate finger target position and finger movement velocities as two stage problem. This design makes learning more efficient.

(+) The experiments are comprehensive. The authors shows how each of the component affects the final performance of the policy.

(+) It also demonstrates the method in the real-world.

**Weaknesses:**

(-) My major concern of this paper is whether the proposed task is more challenging than classical grasping, as claimed by the authors. From my perspective, the grasping process can be roughly divided to 1) hand approaches the object and 2) finger closes. The proposed task use human teleoperation / predefined wrist trajectory for the approaching phase and only learn how / when the finger should grasp the object. In this sense, in terms of task difficulty, what is the difference from firstly moving the wrist to a close-enough position, and then grasp the object under a stationary wrist? Intuitively, I think this is an easier task.

(-) As motivated by my previous argument, there should be more logical arguments on the task difficulty if the author want to emphasize this task “presents a more complex challenge”.

(-) There are formatting issues in particular Table 2.

(-) It relies on perfect point-cloud model.

**Questions:**

As shown in Table 2, stage 1 results are already good if there is no collision. I’m curious if it’s possible to add a collision penalty loss in stage 1 training (similar to the \delta h in RL training) to improve the stage 1 policy?

Is there a more elaborated arguments or evidences that why the proposed task is harder than grasping? On the website, the simulation results look like a classical grasping algorithm while the hand poses are almost the same for all real-world results.

**Limitations:**

This paper is unlikely to have potential negative societal impact.

---

> ### Author Rebuttal · Authors · 2023-08-08
>
> > **Q1: My major concern of this paper is whether the proposed task is more challenging than classical grasping...; As motivated by my previous argument, there should be more logical arguments on the task difficulty...; Is there a more elaborated arguments or evidences that why the proposed task is harder than grasping?**:
>
> Thank you  for bring this up. Due to the page limit, please refer to [Q1 of Common Response](https://openreview.net/forum?id=fwvfxDbUFw&noteId=DtPbwqFxR8).
>
> > **Q2: There are formatting issues in particular Table 2**:
>
> Thank you for your suggestion. We will reexamine the formatting of our paper in the future revision.
>
> > **Q3: It relies on perfect point-cloud model**:
>
> Thank you  for bring this up. Due to the page limit, please refer to [Q4 of Common Response](https://openreview.net/forum?id=fwvfxDbUFw&noteId=DtPbwqFxR8).
>
> > **Q4:  As shown in Table 2, stage 1 results are already good if there is no collision. I’m curious if it’s possible to add a collision penalty loss in stage 1 training (similar to the \delta h in RL training) to improve the stage 1 policy?**:
>
> Thank you for bring this up. We would like to clarify that incorporating a collision penalty loss in stage 1 does have the potential to enhance collision avoidance in grasp pose generation, but it does not specifically address collision avoidance during the grasping procedure. Due to the page limit, please refer to [Q3 of Common Response](https://openreview.net/forum?id=fwvfxDbUFw&noteId=DtPbwqFxR8).
>
> > **Q5: Website simulation results look like a classical grasping algorithm?**:
>
> Thank you for pointing this out. The wrist movement in the website simulation demonstration actually follows a pre-generated human-like trajectory, while the algorithm only controls the finger movement. However, in the classical grasping setting, the algorithm also governs the wrist movement. In order to more effectively illustrate the differentiation between classical grasping and human-assisting grasping, we will render wrist movement trajectories in the revised version of the videos.
>
> > **Q6: While the hand poses are almost the same for all real-world results**:
>
> Thank you for raising this point. We have also observed similar resemblances in grasp poses during our real-world experiments. We hypothesize that this phenomenon could be attributed to the sim2real gap, especially in the context of dynamic aspects. For example, we have discovered that the real hand fails to achieve precise poses when the applied action values fall below a certain threshold, resulting in more similar poses.
>
> However, our policy still demonstrates the capability to achieve successful grasps across various human movement trajectories, indicating the potential of our approach for real-world applications. To effectively showcase the adaptability of our method, we have undertaken other simulation-based experiments, as detailed in [Q2 of Common Response](https://openreview.net/forum?id=fwvfxDbUFw&noteId=DtPbwqFxR8).

---

> > ### Comment · Reviewer_ju1F · 2023-08-16
> >
> > Thank you for your rebuttal. My questions are well-addressed and I will keep my original rating.

---

> > > ### Author Response · Authors · 2023-08-18
> > > **Thank You!**
> > >
> > > We are so glad that our responses help address your concerns. Thanks again for all your valuable feedback!

---

### Author Rebuttal · Authors · 2023-08-08

## **Common Response**:
We thank all reviewers for appreciating our ideas and experiments. “A unique dexterous grasp task **(PG5Z)**". "The proposed framework is intuitive and is properly designed for the challenges of its task **(GtH3)**". "Formulate the learning from a set of successful grasps as a denoising problem, which is quite interesting and novel **(ju1F)**".  "Demonstrated strong performance in both simulated and real-world experiments **(GtH3)**". "Showcasing the practical applicability of the proposed method **(KjWd)**".

However, we notice that some reviewers **(PG5Z, KjWd, ju1F)** might confused about the **(Q1) Distinction between Teleoperation, Automatic Dexterous Grasping (Classical Grasping),  and Human-Assisting Grasping**.

The main motivation and practical application **(PG5Z, KjWd, ju1F)** of assisting grasping is assisting upper limb amputees with prosthetic hands, as show in **Figure 1 of main pdf**.

 - **Assisting Grasping v.s. Teleoperation.**
Traditional teleoperation method[1] **(PG5Z, KjWd)** is unsuitable for assisting grasping, because we can not get information of human finger. In real-world experiments, we use teleoperation solely to get human wrist poses for mimicking grasping by upper limb amputees.

 - **Assisting Grasping v.s. Automatic Dexterous Grasping .** Compared to automatic dexterous grasping **(KjWd, ju1F)**, agent can not control the wrist of the prosthetic hand in assisting grasping, which poses challenge for user-aware grasping. Due to the complex and diverse behaviours when human controls the wrist, deciding how and when to grasp is challenging for agent without considering the human movement. For instance, if the human first moves the wrist close to the object and then grasp, the agent still must factor in the object-wrist relationship for how to grasp. On the other side, grasping with a fixed wrist pose instead of close the finger in advance can result in table obstructs finger closure, requiring humans to adjust their wrist. This affects grasp fluency and introduces burden, a crucial consideration for humans. Moreover, humans may face dynamic objects, as shown in **Section 1 of the anonymous project page**. Ignoring human movement could result in failed grasps.

We also noticed that some reviewers concerned about **Q2: The Necessity of Adaptability, and the Adaptability of Our Proposed Method (KjWd, GtH3, ju1F)**.

 - **Necessity of Adaptability** It's crucial to grasp different parts **(KjWd)** of objects in daily life, as many objects serve multiple purposes. For example, a hammer can be used for nailing and sweeping. Adaptability also requires for same purpose. When cleaning shoes, human need to hold different parts of shoes. Moreover, objects may be put in different places by various poses, which makes grasping a specific part challenging.

 - **Adaptability of Our Proposed Method** **(KjWd, GtH3, ju1F)** Quantitatively, we've evaluated "posture" measuring alignment between ours and intended grasp poses, as shown in **Figure 4 of the main pdf**. on the other side, as  shown in **Table 1 of the rebuttal pdf**, our method excels at grasping various parts of objects. Qualitatively, **Figure 1 of the rebuttal pdf**  shows human's intended grasp and our method's close-to-ground-truth pose. In another experiment, as shown in **Section 1 of the anonymous project page**, GraspGF produces diverse grasp poses as the wrist moves.

We futher noticed that some reviewers **(KjWd, ju1F)** might confused about **Q3: Distinction Between Collision in Grasp Pose Generation and Collision in Grasping Procedure**.

There are  two collision types in our current problem.

 - **Collision in Grasp Pose Generation**  The first arises during grasp pose generation, e.g., "how to grasp". Generated grasp poses can sometimes result in object collisions. To address this, we can heed the reviewer's advice **(ju1F)** to include a collision penalty, potentially improve the primitive policy's performance.

 - **Collision in Grasping Procedure** This collision relates to "when to grasp" regardless of direct object collision in the generated pose. Even if a pose has no collision with objects, collisions can still occur when the agent control fingers to reach the desired pose—colliding with the table or object itself.

 - **Necessity of Residual Policy** To clarify, results of "$a^p$ w/o coll" **(KjWd, ju1F)** are obtained by disabling collisions in grasping procedure. **Figure 2 of the rebuttal pdf** highlights that while the final grasp pose can grasp the object, poses are achieved before the agent intends to grasp. This shows a residual policy **(KjWd)**  is crucial to decide the ideal grasp timing.

Some reviewers **(ju1F, GtH3)** also highlights **Q4: Limitation of Perfect and Full Point Cloud**.

We have discussed in **Section 7 of the main pdf**. Addressing this concern can involve using teacher-student learning[2] or adapting our pipeline to be trained with partial point cloud input **(ju1F, GtH3)**. For handling the gap between real point cloud and sim point cloud, we may add noises[3] to point cloud observations for training **(ju1F)**.

We sincerely hope our work contributes to the Machine Learning + Robotics research community and eventually improve the social welfare. Below we reply to reviewers’ questions point-by-point. Thanks again for your valuable comments and suggestions!

**Reference**:

[1] Handa, Ankur, et al. "Dexpilot: Vision-based teleoperation of a dexterous robotic hand-arm system."

[2] Chen, Tao, Jie Xu, and Pulkit Agrawal. "A system for general in-hand object re-orientation."

[3] Dai, Qiyu, et al. "Domain randomization-enhanced depth simulation and restoration for perceiving and grasping specular and transparent objects."

---

### Decision · Program_Chairs · 2023-09-21

**Decision:**

Accept (poster)

**Comment:**

The paper proposes a novel approach to fuse machine learning of motor skills with human input through tele-operation.

The reviewers found the method sound and novel, and the experimental results very convincing. They initially had questions about some details in the setting, the method, and assumptions/limitations. Some shortcomings in the presentation were pointed out.

After the rebuttal the main concerns of the reviewers have been successfully addressed and they are all vote in favor of accepting the paper.